# Unexplored diversity and potential functions of extra-chromosomal elements

Haoyu Liu,[1] Jia Sun,[1] JuanJuan Si,[1] Yi Liao,[1] Jiaqi Bai,[1] Xia Li,[2] Limin Wang,[3] Kuojun Cai,[4] Wei Ni,[1] Ping Zhou,[3] Shengwei Hu[1]

**ABSTRACT** In addition to the main chromosome, approximately 10% bacterial genomes have one or more large secondary replicons, including a unique type of replicon known as "chromid," which has plasmid-type replication and partition systems but carries core genes. Their nucleotide composition of chromids is very similar to that of their related chromosomes. However, the distribution, characteristics, functions, and origins of these chromids remain unclear. In this study, we established a workflow to identify chromids from complete bacterial genomes and screened 36,322 complete bacterial genomes, resulting in the identification of 1,104 bacterial genomes with chromids as secondary replicons. These chromid-carrying bacteria belong to eight phyla and 73 genera, exhibiting diversity and a wide global distribution. We analyzed the characteristics of chromids and found that their average size is larger than that of "megaplasmids" and that multi-chromid bacteria exist. Furthermore, chromids encode genes related to bacterial respiratory chain enzyme complexes and antiviral systems, expanding bacterial metabolic capabilities and enhancing their antiviral defenses. In addition, we developed an automated identification program, Chromid-Finder, for identifying chromid sequences in metagenomic data, which has demonstrated outstanding performance. To demonstrate its application, we analyzed 92,143 metagenome-assembled genomes (MAGs) from the human gut microbiome. We found that the distribution of chromid-carrying bacteria in the human gut is closely associated with host age, health status, and geographic location. Species with chromids exhibit unique functional capabilities, showing good separation at the phylum level.

**IMPORTANCE** In this study, we have developed a workflow to identify chromids from complete bacterial genomes. We utilized this workflow to search for chromids in the latest NCBI RefSeq databases, to map the distribution of bacteria carrying chromids, to identify the characteristics of bacterial chromids, to discuss their origins, and to investigate their roles in bacterial life. To address the growing volume of metagenomic data, we developed a high-performance automated identification program, Chromid-Finder, designed to identify chromids and their corresponding bacterial main chromosomes within extensive metagenomic data sets. Using this tool, we analyzed 92,143 metagenome-assembled genomes (MAGs) from the human gut microbiome.

**KEYWORDS** chromid, bacterial respiratory chain, antiviral systems, chromid-finder tool

Since John Cairns first reported the autoradiographic images of *Escherichia coli* DNA in 1963, demonstrating that the *E. coli* genome consists of a single circular chromosome (1), it has been widely believed that all bacterial genomes consist of a single circular chromosome, possibly accompanied by some smaller, non-essential circular plasmids. However, as research progressed, the discovery of linear plasmids (2) and linear chromosomes (3) gradually began to change this perspective. The emergence

Address correspondence to Wei Ni, niweiwonderful@sina.com, Ping Zhou, zhpxqf@163.com, or Shengwei Hu, hushengwei@163.com.

Haoyu Liu, Jia Sun, and JuanJuan Si contributed equally to this article. Author order was determined by minor differences in their contributions during manuscript preparation.

The authors declare no conflict of interest.

See the funding table on p. 18.

of megaplasmids (4) further challenged the notion that the entire bacterial genome is located on the chromosome.

In addition to the main chromosome, approximately 10% bacterial genomes have one or more large secondary replicons (any replicon that is not the bacterial chromosome) (5). In another study, a different type of secondary replicon was identified, distinct from plasmids or megaplasmids: the chromid (6). Chromids are defined by the following characteristics: (i) they have plasmid-type replication and partitioning systems; (ii) their nucleotide composition is very similar to that of the chromosomes they are associated with; and (iii) they carry core genes that are found on the chromosomes of other species.

Due to technology limitations at the time, neither the study that first introduced the concept of chromids in 2010 (which manually screened 799 complete bacterial genomes) (6), nor a later review in 2017 (which had a rough screening of 4,541 complete bacterial genomes) (5), provided a comprehensive and accurate workflow for the large-scale screening of chromids across bacterial genomes. As a result, despite the identification of chromids in bacteria and archaea such as *Haloferax volcanii* (7), *Novosphingobium terrae* (8), and *Plesiomonas shigelloides* (9), where they play crucial roles in bacterial survival, large-scale, systematic identification and analysis of chromids remains a challenge.

In this study, we have developed a workflow to identify chromids from complete bacterial genomes. We utilized this workflow to search for chromids in the latest NCBI RefSeq databases (10), to map the distribution of bacteria carrying chromids, to identify the characteristics of bacterial chromids, to discuss their origins, and to investigate their roles in bacterial life.

To address the growing volume of metagenomic data, we developed a high-performance automated identification program, Chromid-Finder, designed to identify chromids and their corresponding bacterial main chromosomes within extensive metagenomic data sets. Using this tool, we analyzed 92,143 metagenome-assembled genomes (MAGs) from the human gut microbiome (11). Chromid-Finder is freely available for download at https://github.com/China-LHY/Chromid-Finder.

## RESULTS

### Chromid-carrying bacteria are found in various types of bacteria across the globe

We downloaded 36,322 complete bacterial genomes, representing 6,295 species, from the NCBI RefSeq database. Since no publicly available methods for identifying chromids existed, we developed a workflow to identify chromids in complete bacterial genomes based on the previously proposed chromid concept (6). As shown in Fig. 1a, we identified 1,104 chromid-carrying bacterial genomes (Table S1), representing 247 species. These 1,104 chromid-carrying bacteria are distributed across eight phyla (*Acidobacteriota, Bacillota, Bacteroidota, Cyanobacteriota, Deinococcota, Fusobacteriota, Pseudomonadota,* and *Spirochaetota*), with a notable concentration in the *Pseudomonadota* phylum. The chromosomes of these bacteria were validated through multiple sequence alignment using GTDB-Tk, and a phylogenetic tree was constructed (Fig. 1b).

The size of the chromosomes in chromid-carrying bacteria ranged from 1.27 Mb to 9.04 Mb, with an average of 3.41 Mb and a median of 3.51 Mb. In contrast, an initial screening of 17,236 bacterial genomes with two or more nucleotide sequences revealed chromosome sizes ranging from 12.35 Mb to 0.55 Mb, with an average of 4.22 Mb and a median of 4.61 Mb. These results clearly indicate that the average chromosome size in chromid-carrying bacteria is smaller than that in bacteria with plasmids or megaplasmids.

Based on the constructed classification Sankey plot (Fig. 2), we analyzed the taxonomic distribution of bacteria carrying chromids. These bacteria are broadly distributed across 8 phyla, 14 classes, 31 orders, 44 families, and 73 genera. Previous studies have shown that chromids are rich in genus-specific genes and may contribute to the formation of new genera (6). Accordingly, our classification extends to the genus level.

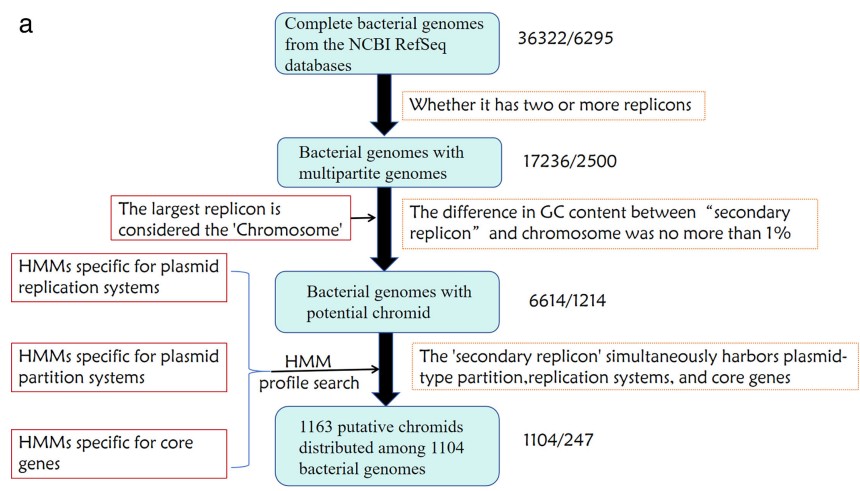

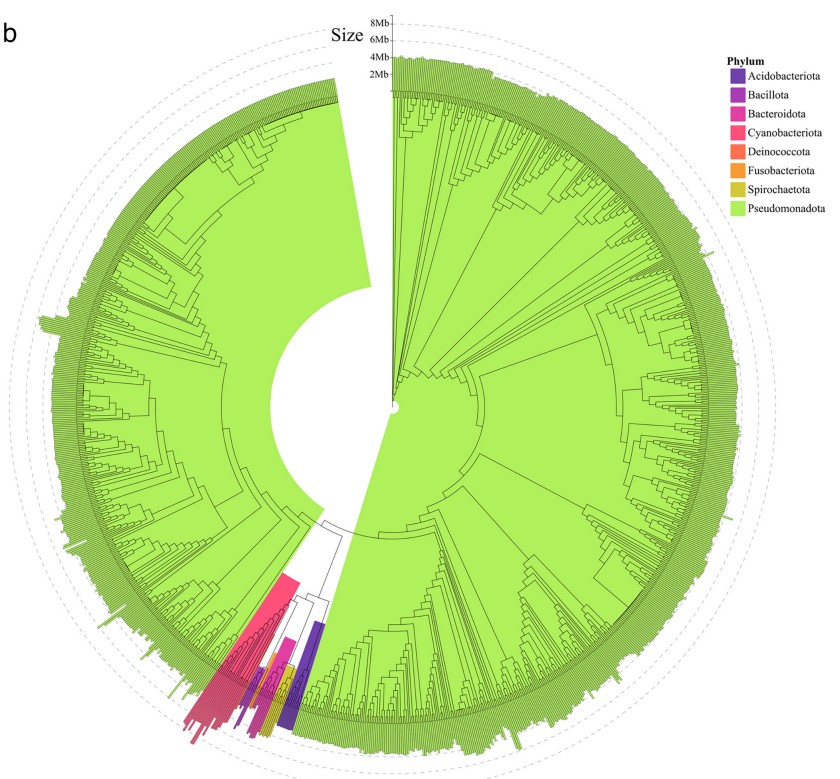

**FIG 1** Diversity of chromid-carrying bacteria. (a) The method for screening chromids. The numbers represent the quantity of bacterial genomes/species. (b) Phylogenetic tree of the chromosomes from chromid-carrying bacteria.

The top five genera with the highest number of chromid-carrying bacteria are *Burkholderia* (n = 441), *Brucella* (n = 204), *Agrobacterium* (n = 91), *Vibrio* (n = 88), and *Paraburkholderia* (n = 55). Subsequently, we calculated the ratio of chromid-carrying strain genomes to the total number of complete genomes sequenced within each bacterial genus (Table 1; the complete table is provided in Table S2). These genera include bacteria that have significant impacts on human productivity and daily life.

Chromid-carrying bacteria have been identified in diverse geographical locations and ecosystems worldwide, including host-associated, aquatic, terrestrial, and sediment biomes (Fig. 3a). A country-level analysis revealed their presence across all seven continents, four oceans (Fig. 3b), and 71 countries (Fig. S2). This widespread distribution

TABLE 1 The ratio of chromid-carrying strain genomes to the total number of complete genomes sequenced within each bacterial genus (only bacteria genera whose genome number is not 1 and ratio ≥90% are shown)

| Genus | Chromid-carrying | All | Ratio (%) |
|---|---|---|---|
| *Flammeovirga* | 4 | 4 | 100 |
| *Trichormus* | 2 | 2 | 100 |
| *Aminobacter* | 8 | 8 | 100 |
| *Cereibacter* | 10 | 10 | 100 |
| *Azospirillum* | 17 | 17 | 100 |
| *Caballeronia* | 10 | 10 | 100 |
| *Agrobacterium* | 91 | 94 | 96.81 |
| *Brucella* | 204 | 212 | 96.23 |
| *Paraburkholderia* | 47 | 49 | 95.92 |
| *Burkholderia* | 428 | 450 | 95.11 |
| *Chloracidobacterium* | 9 | 10 | 90 |

highlights their ability to inhabit an extraordinary variety of human and natural environments, underscoring their extensive global prevalence.

## Characteristics of bacterial chromids: larger than megaplasmids, with some bacteria carrying multiple chromids

Although most chromid-carrying bacteria possess only a single chromid, a small subset harbors two or more. As shown in Fig. 4a, our results reveal that *Pseudanabaena galeata* from the *Cyanobacteriota* phylum carries up to four chromids. Notably, even within the same genus or species, the number of chromids can vary across individual bacterial genomes. For instance, in the *Azospirillum* genus, the number of chromids ranges from one to three within a single genome. Specifically, *Azospirillum brasilense* can contain two to three chromids along with several plasmid/megaplasmid sequences. Similarly, in the

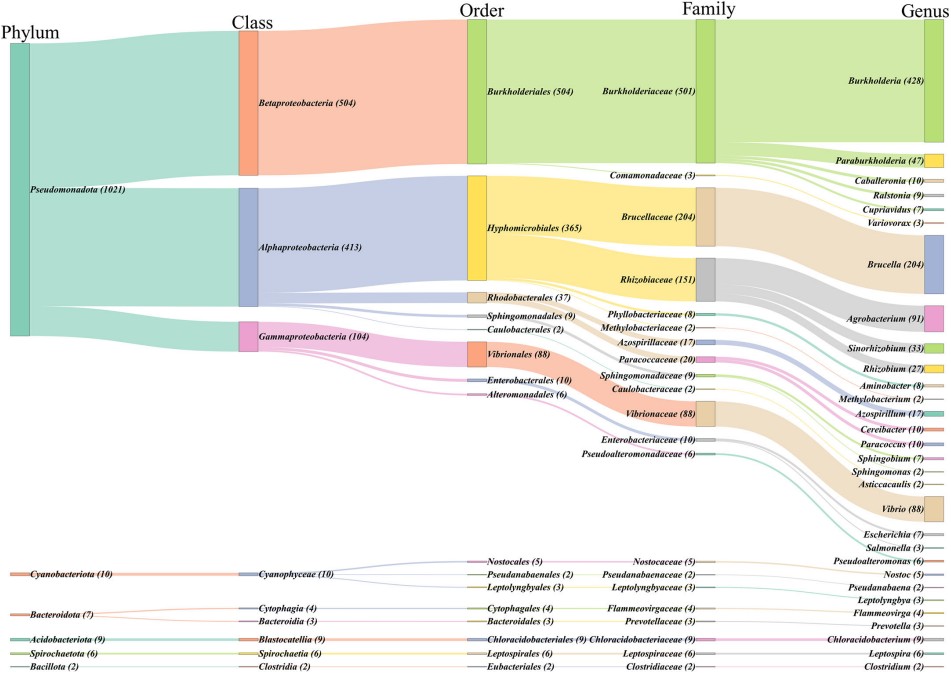

FIG 2 Phylogenetic distribution of chromid-carrying bacteria across different taxonomic levels. The numbers in parentheses indicate the quantity of chromid-carrying bacteria within each specific taxonomic unit. Only display the genera with the number of chromid-carrying bacterial genomes ≥3 or containing ≥2 species (*n* = 30). The complete Sankey diagram is provided in Fig. S1.

**TABLE 2** Chromid screening results for est data set 2 (tetranucleotide relative abundance distance <1.6[a]

| Prediction | Verified present (T) | Verified absent (F) | Total |
|---|---|---|---|
| Predicted present (P) | True positive (TP): 33 | False positive (FP): 0 | P (total predicted positive): 33 |
| Predicted absent (N) | False negative (FN): 12 | True negative (TN): 111 | N (total predicted negative): 123 |
| Total | T (total verified positive): 45 | F (total verified negative): 111 | Total samples (P + N or T + F): 156 |

[a]Accuracy (Acc) = 92.31%, precision = 100%, true positive rate (TPR) = 73.33%, false positive rate (FPR) = 0%.

genus *Cereibacter*, the species *Cereibacter sphaeroides* can have one to two chromids per genome. This variation may represent an evolutionary continuum, showcasing the transition of plasmids into chromids in these bacteria.

Figure 4b displays a boxplot illustrating the sizes of chromids categorized by phylum. The boxplot reveals that *Pseudomonadota* possess relatively large genomes, whereas *Cyanobacteriota* exhibit smaller genomes. And *Bacillota* contain some exceptionally small genomes, which may reflect their adaptability or lifestyle. The histogram shows that chromid genome sizes range from 0.04 to 4.00 Mb, with an average size of 1.98 Mb (Fig. S3a). Various thresholds have been proposed to define the size of megaplasmids. For example, diCenzo and Finan's seminal review on multipartite genomes set the threshold at 0.35 Mb, based on 10% of the median bacterial genome size (5). Other studies have used a threshold of 0.1 Mb (12), while a more reasonable approach might link plasmid size to the size of other replicons in the bacterial genome: a size greater than or equal to 5% of the total genome size (13). In summary, regardless of the threshold used, the average size of chromids surpasses these criteria.

We constructed a heatmap to illustrate the distribution of Rep genes, Par genes, core genes, tRNA genes, and rRNA genes, followed by complete linkage clustering analysis (Fig. S3b). Meanwhile, the clustering results of the heatmap revealed that tRNA genes exhibited correlations with rRNA genes, while Rep protein genes showed associations with Par protein genes, which is consistent with our screening logic.

Our analysis of the differences in GC content between chromids and their corresponding chromosomes (Fig. S4a) revealed a distribution pattern resembling a normal curve. This indicates that our data set is well-balanced and supports the stability and reliability of our selection criteria. In a review, diCenzo and Finan proposed using the dinucleotide relative abundance distance between chromids and chromosomes as a preliminary screening criterion, recommending a threshold of ≤0.4 (5). However, our bar chart (Fig. S4b) suggests that while this threshold effectively identifies most chromids, it may exclude certain candidates.

## Chromids as key players in bacterial respiratory chains

Bacteria possess remarkable energy metabolism capabilities, enabling them to survive in diverse and extreme environments, from acidic ponds and hot springs to the anaerobic guts of animals. Their exceptional adaptability is reflected at the molecular level in the modular structure of their respiratory chains, which are composed of enzyme complexes. Referring to Kaila's study (14), we mapped the respiratory chain enzyme complexes and electron transfer pathways in *Burkholderia pseudomallei* and discovered that proteins encoded on chromids contribute to the composition of these complexes (Fig. 5a).

While the core principles of the respiratory chain are highly conserved across all domains of life, bacterial respiratory chains exhibit significant variability. To illustrate the role of chromids in bacterial respiratory processes, we analyzed several representative species (Fig. 5b). We selected five species: *Azospirillum brasilense*, *Burkholderia pseudomallei*, *Paracoccus versutus*, *Vibrio chagasii*, and *Vibrio cholerae,* to represent the three distinct roles of chromids in bacterial respiratory chains.

i. No involvement in the bacterial respiratory chain (144/1,163), such as chromid-3 in *Azospirillum brasilense* and the chromid in *Vibrio chagasii*.

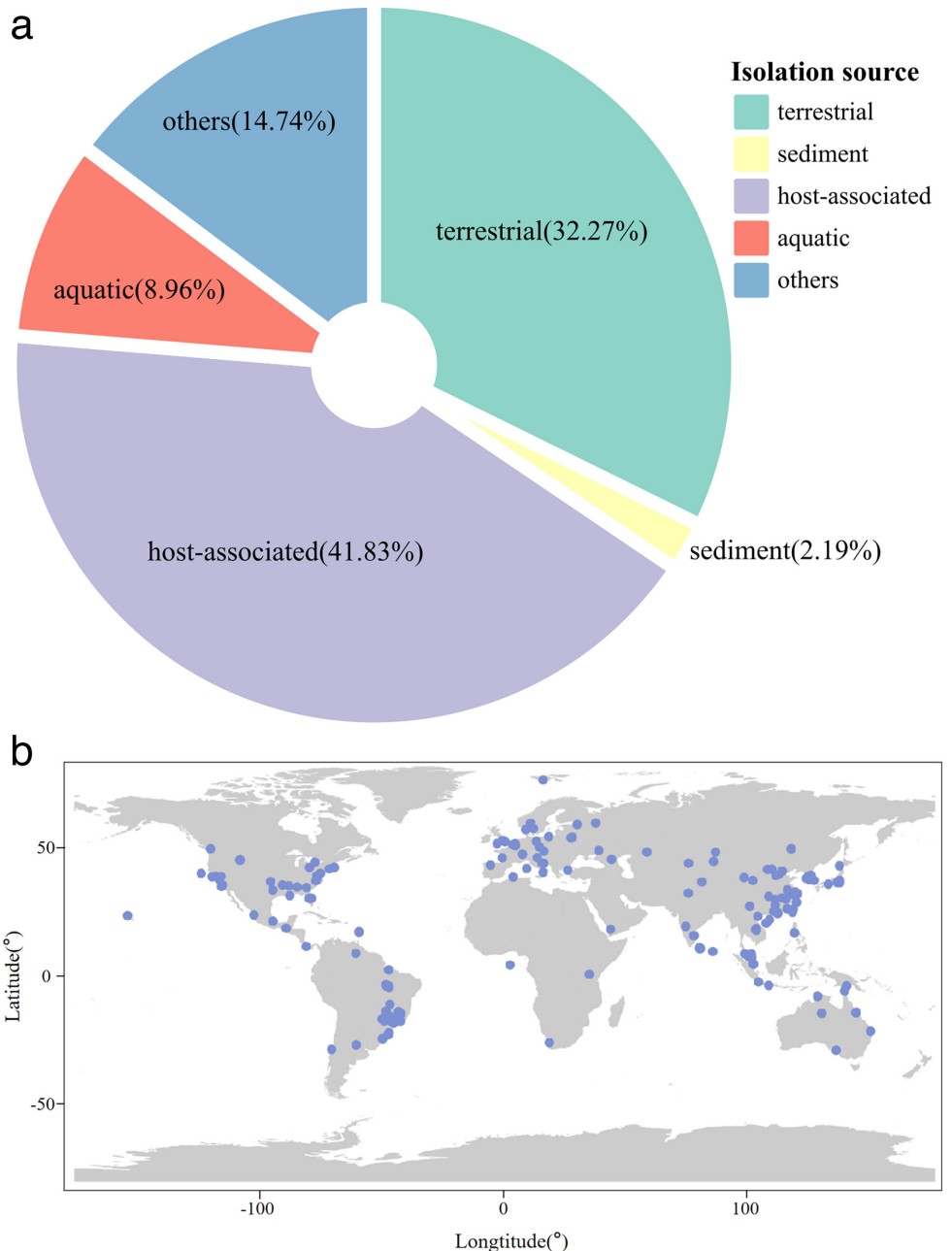

**FIG 3** Global distribution and isolation sources of chromid-carrying bacteria. (a) Based on the isolation source data of the available chromid-carrying bacterial genomes ($n = 502$), the sample sources were roughly divided into four categories: terrestrial, sediment, host-associated, aquatic, and others. (b) Geographical distribution of chromids. Each point represents a geographic location, showing only chromid-carrying bacterial genomes with available geographic coordinates ($n = 293$).

 ii. Independent encoding of all protein genes for a respiratory chain enzyme complex (521/1,163), such as chromid-1 in *Azospirillum brasilense* and the chromid in *Burkholderia pseudomallei*.

 iii. Supporting the bacterial chromosome in synthesizing respiratory chain enzyme complexes (489/1,163), such as chromid-2 in *Azospirillum brasilense*, the chromid in *Paracoccus versutus*, and the chromid in *Vibrio chagasii*.

 Although this small selection of species cannot represent the full diversity of bacterial respiratory chains, it highlights the critical roles chromids play in certain bacteria, either

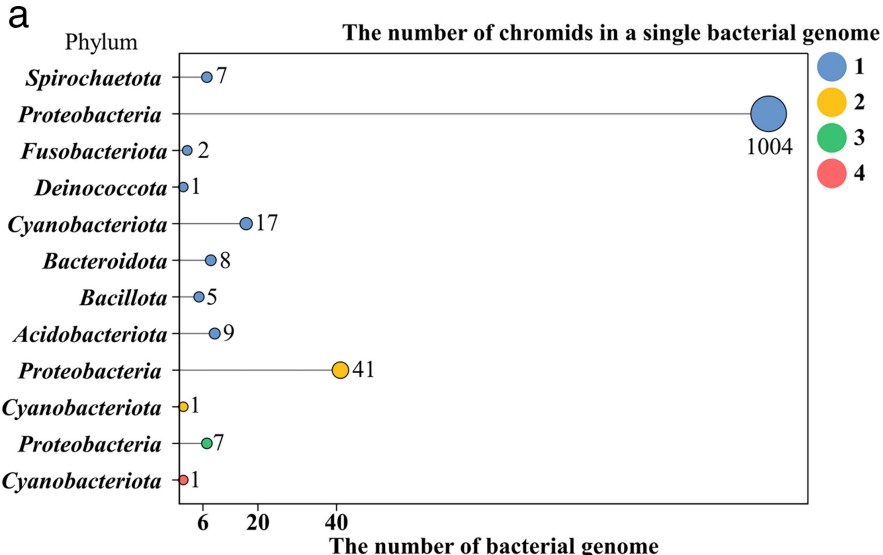

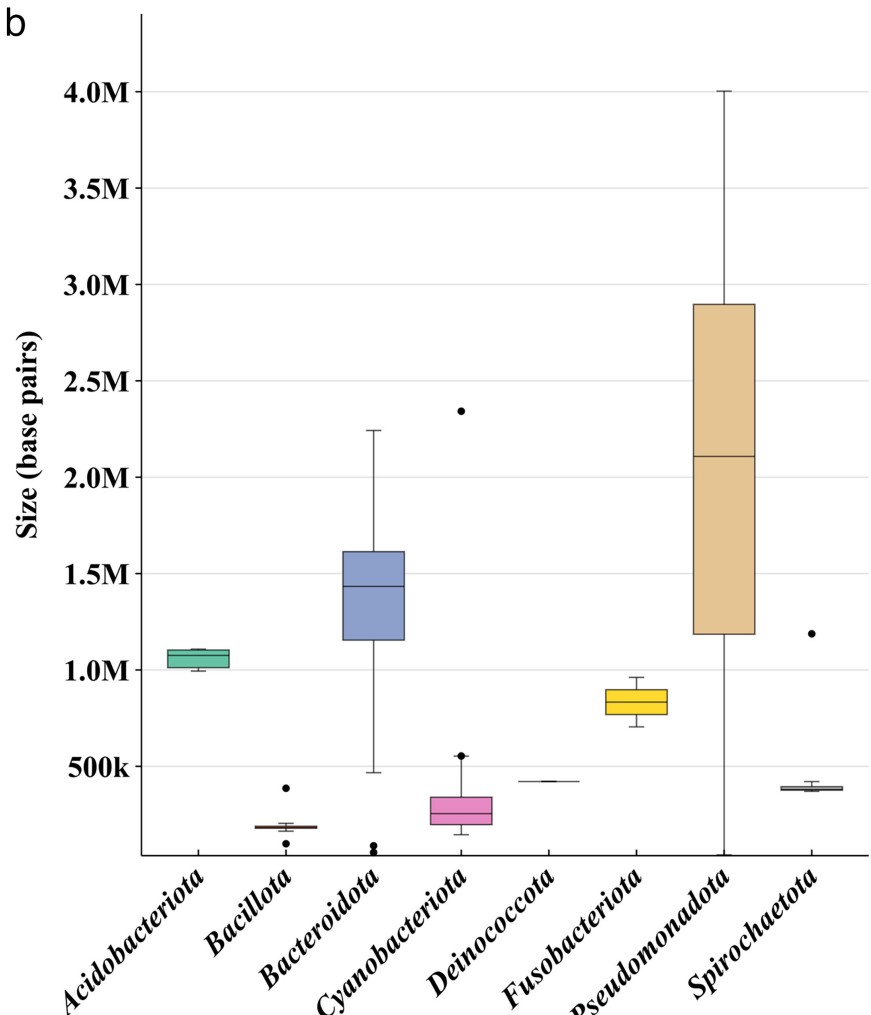

**FIG 4**   Characteristics of bacterial chromids. (a) The distribution of chromids within individual bacterial genomes. (b) Box plot showing the distribution of chromid sizes by phylum.

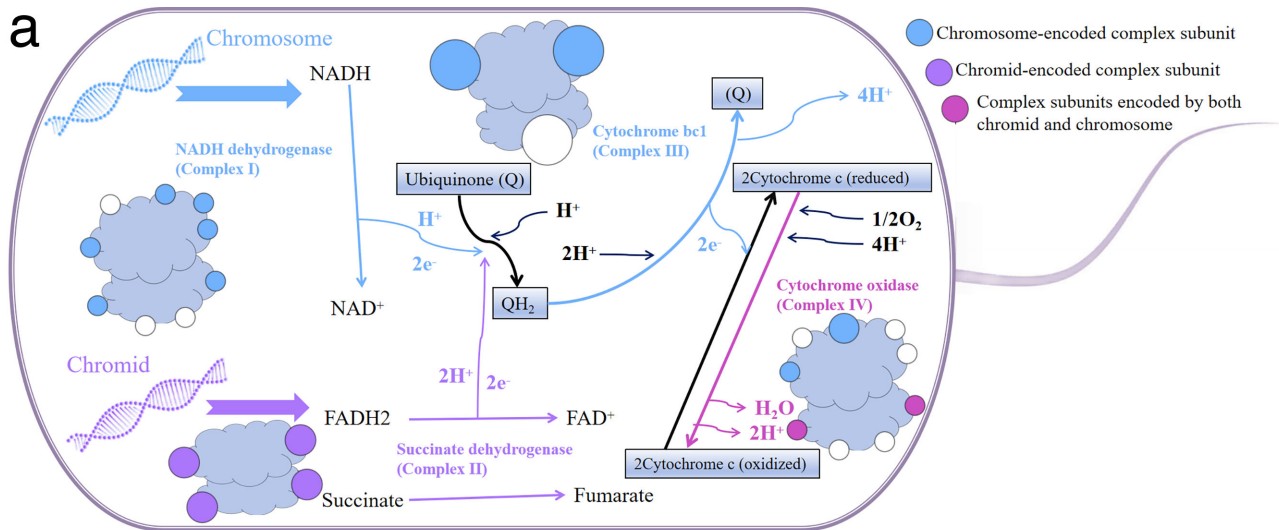

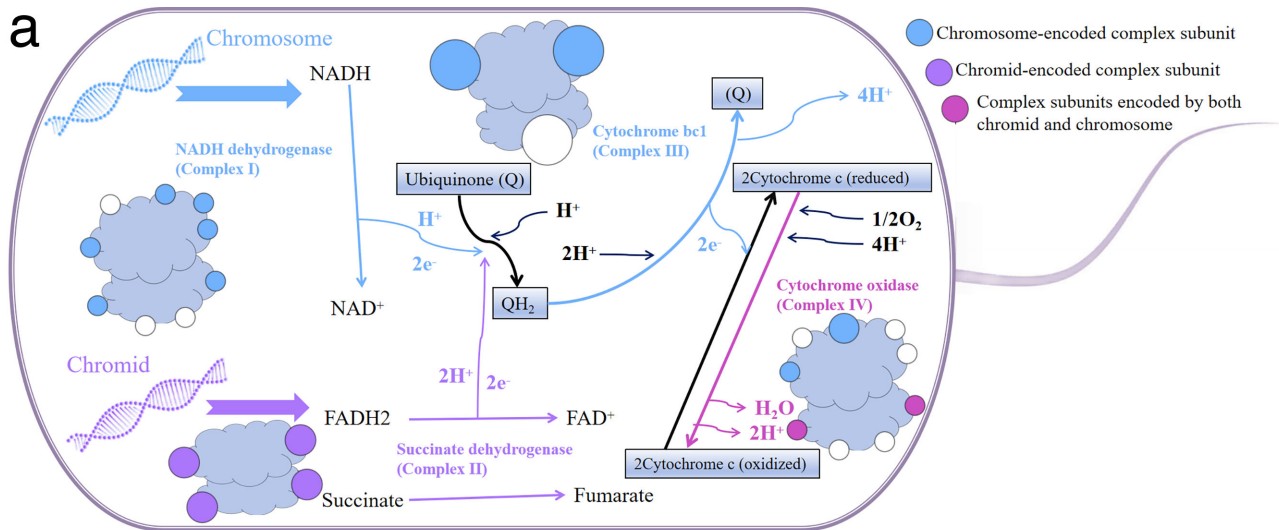

**FIG 5** The role of chromids in the bacterial respiratory chain. (a) Model of the *Burkholderia pseudomallei* respiratory chain. Proteins encoded by both the chromosome and chromids can participate in the bacterial respiratory chain. (b) Binary representation of proteins related to respiratory chain enzyme complexes encoded by different replicons in representative bacterial genomes. NDUFAF: NADH dehydrogenase 1 alpha subcomplex assembly factor; NDUFS: NADH dehydrogenase Fe-S protein; frdA: succinate dehydrogenase flavoprotein subunit; frdB: succinate dehydrogenase iron-sulfur subunit; frdC: succinate dehydrogenase subunit C, frdD: succinate dehydrogenase subunit D, petB: ubiquinol-cytochrome c reductase cytochrome b subunit; petC: ubiquinol-cytochrome c reductase cytochrome c1 subunit; fbcH: ubiquinol-cytochrome c reductase cytochrome b/c1 subunit; coxA/B/C/D: cytochrome c oxidase subunit I/II/III/IV; ccoN/O/P/Q: cytochrome c oxidase cbb3-type subunit I/II/III/IV; COX11: cytochrome c oxidase assembly protein subunit 11 (simultaneous encoding of petB/C or fbcH can function as Complex III).

supplementing or performing indispensable functions in the respiratory process. This finding is particularly exciting as it underscores the essential contributions of chromids to fundamental bacterial life processes.

## Chromids as critical players in bacterial antiviral defense mechanisms

We analyzed the types and numbers of antiviral defense genes across all replicons in chromid-carrying bacteria: those located on the chromosome, the chromids, and on

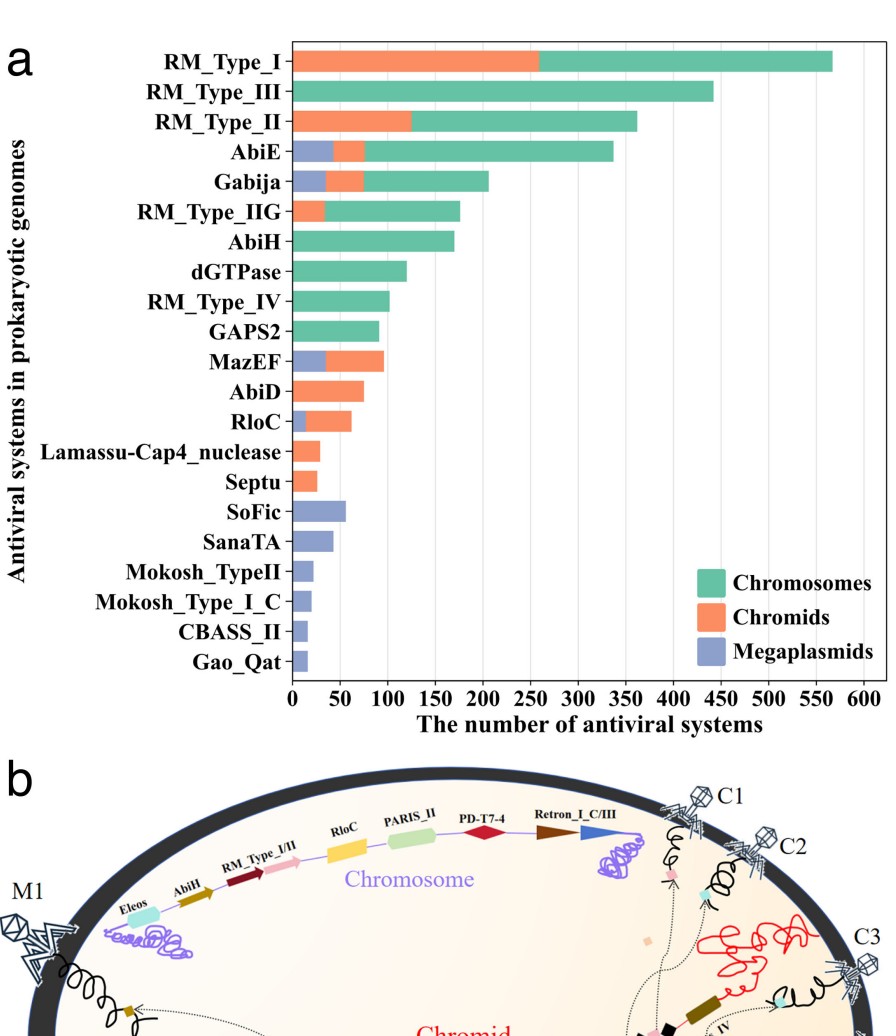

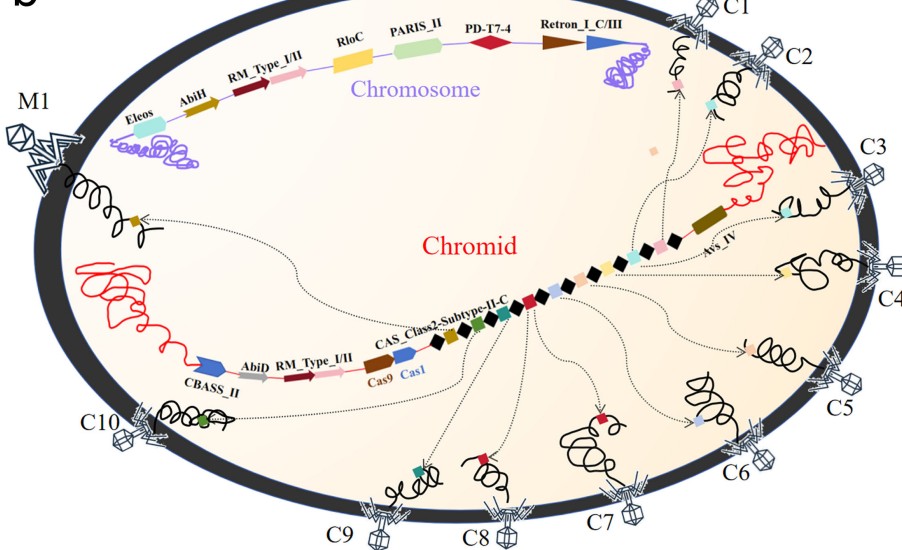

**FIG 6** Chromids enhance bacterial antiviral systems. (a) Distribution of bacterial antiviral systems across chromosomes and chromids. Only display separately the top ten antiviral defense systems in number on bacterial chromosomes and chromids. (b) Distribution of antiviral systems in *Prevotella fusca* and the cellular diagram of CRISPR-targeted bacteria-phage interactions. Arrows indicate CRISPR-Cas targeting of phages and phage genomes. C, Caudoviricetes; M, Microviridae.

megaplasmids. For the megaplasmids, we followed the standard proposed by George C. diCenzo et al. (defining megaplasmids as ≥350 kb), downloading and analyzing 472 megaplasmids originating from 1,104 chromid-carrying bacterial genomes. The distribution of some antiviral systems is shown in the bar chart in Fig. 6a. A simple method was used to compare the number of antiviral systems encoded on bacterial chromosomes and chromids: specifically, the total number of antiviral systems was divided by the total sequence lengths of bacterial chromosomes and chromids, respectively. This yielded a density of 1.147 antiviral systems per Mbp on chromosomes,

1.296 antiviral systems per Mbp on megaplasmids, and 0.644 antiviral systems per Mbp on chromids. This indicates that the number of antiviral system genes encoded on chromids is considerable, and their distribution differs from that on bacterial chromosomes and megaplasmids. This suggests that chromids can supplement bacterial antiviral systems, thereby enhancing the bacteria's defense against phages.

To further illustrate the role of chromids in bacterial antiviral defense, we examined *Prevotella fusca* and created a cellular diagram (Fig. 6b). In *P. fusca*, the chromid contains a CRISPR-Cas system, whereas the chromosome neither has CRISPR array sequences nor encodes Cas proteins. We detailed the bacterial defense systems and identified the potential phage targets of the CRISPR-Cas system. The chromid contains 11 spacers, primarily targeting two types of phages: *Caudoviricetes* and *Microviridae*. It encodes the Cas1 and Cas9 proteins, characteristic of the type II-C CRISPR system (15). Notably, the defense systems on the chromosome and chromid of *Prevotella fusca* are different, indicating that the chromid plays a critical role in supplementing the bacterial defense system and enhancing the bacterium's ability to defend against phage attacks.

## Develop and test Chromid-Finder

The workflow of Chromid-Finder consists of four main steps (Fig. 7). In summary, the process begins with the input of assembled MAGs, followed by the calculation of GC

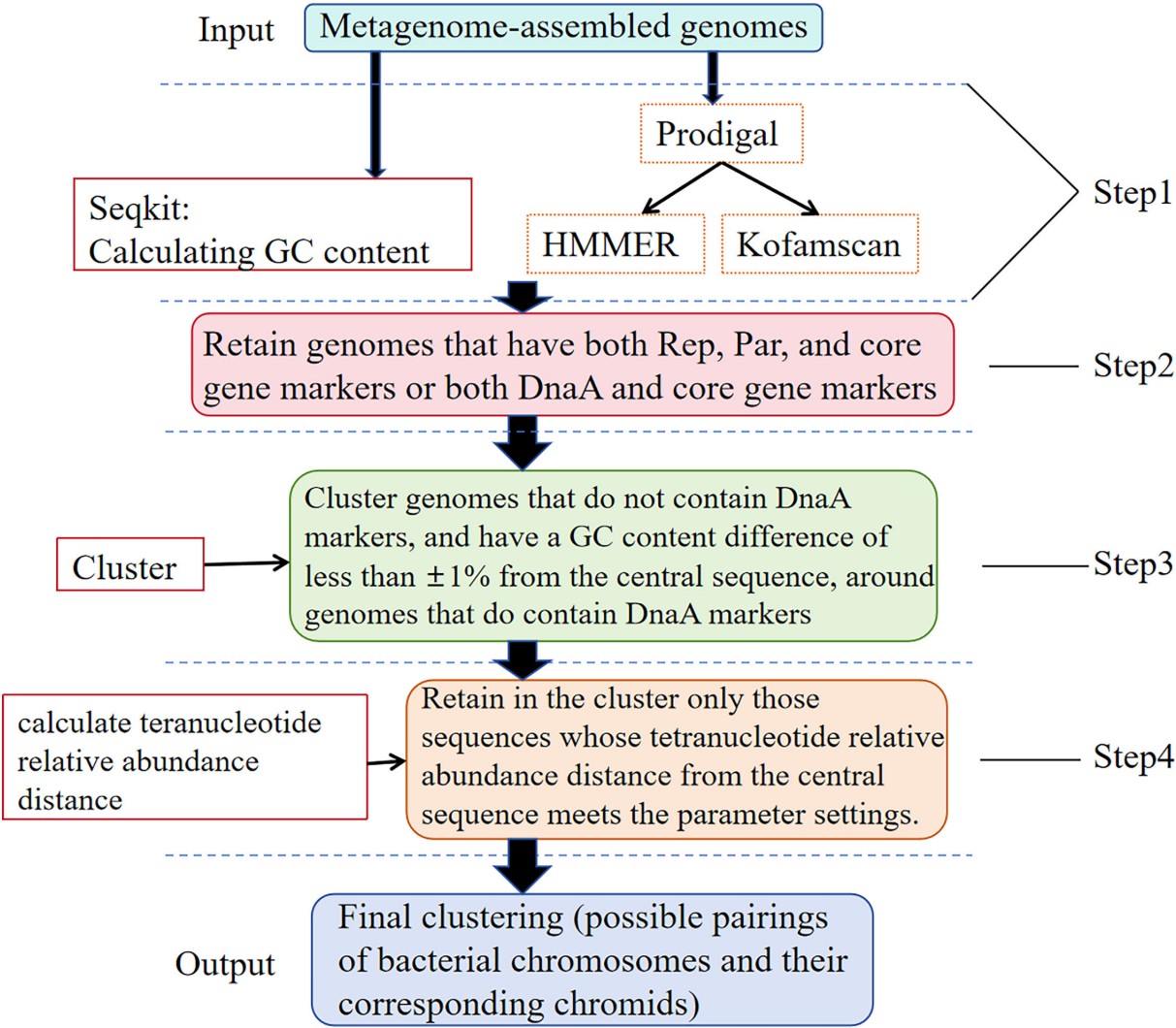

**FIG 7** Workflow of Chromid-Finder.

content, alignment, and statistical analysis of marker proteins to retain gene sequences that meet specific criteria. Next, an initial clustering of the remaining genomes is performed, using genomes containing DnaA and its homologs as clustering centers. This method relies on DnaA-dependent replication initiation, which is considered unique to bacterial chromosomes (16). Clusters are formed from sequences shorter than the center and with GC content differences within ±1%. Subsequently, the tetranucleotide relative abundance distance is calculated for each sequence in the cluster compared to the center sequence, retaining those that conform to the parameter settings. Finally, the output consists of the final clustering results, where other sequences in the cluster are identified as potential chromids, and the center sequence corresponds to the bacterial chromosome associated with the chromid.

To evaluate the performance of the Chromid-Finder software, we input 62,850 sequences from 36,322 complete bacterial genomes in the NCBI RefSeq database, treating them as 62,850 independent genomic sequences by removing their corresponding relationships. We confirmed using test dataset 1 that the software correctly implements the required steps. Furthermore, compared to our initial workflow, our tool has relaxed the filtering criteria for Rep proteins, Par proteins, and core genes and additionally incorporated hidden Markov model (HMM) profiles related to bacterial chromosome replication origin specificity and the parameter of tetranucleotide relative abundance distance. This was done with the aim of testing whether the software can effectively function as a screening tool when faced with numerous independent replicon sequence inputs, rather than when comparing different replicons within the same genome. The results demonstrate that at a tetranucleotide relative abundance distance threshold of 1.6, we identified 1,043 chromids out of 1,163 putative chromids in test dataset 1, achieving a recall rate of 89.68%.

Additionally, we tested 156 sequences from 39 bacterial genomes cited in 28 publications, again removing their corresponding relationships and treating them as 156 independent genomic sequences input into Chromid-Finder. We plotted receiver operating characteristic (ROC) curves, precision-recall (PR) curves, and parameter sensitivity analysis curves centered on the tetranucleotide relative abundance parameter. Although the test data set had a relatively small sample size, it demonstrated excellent AUC values in the ROC and PR curves, while the parameter sensitivity analysis also showed robust performance (Fig. 8). Based on Fig. 8 and Fig. S5, the tetranucleotide relative abundance parameter for our subsequent experiments was set to 1.6. The output results are shown in Table 2, where TP represents sequences predicted as chromids that are indeed chromids. We also conducted a detailed analysis of the identification process (see Table S4). In summary, out of 45 chromids across these 39 bacterial genomes, we accurately identified 33 chromids (32+1). Among the remaining 12 chromids that were not identified: 10 lacked a core gene, 7 lacked a Rep protein, and 5 had a difference in GC content exceeding 1%. Furthermore, regarding GCF_000730165.1, the cited article

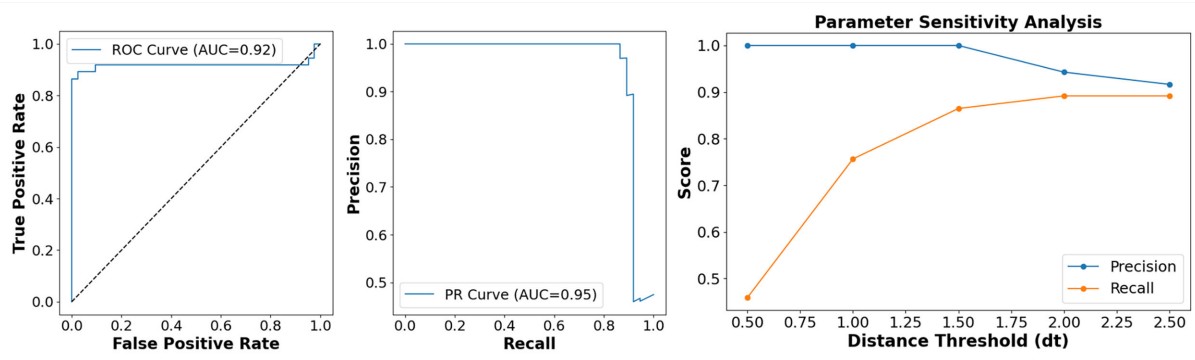

**FIG 8** The ROC curves, precision-recall plots, and parameter sensitivity analyses for the test data set 2 (tetranucleotide relative abundance distance). The parameters only come into play at step 4 of the Chromid-Finder workflow.

(17) asserts that there are three chromosomes in this genome, which conflicts with our definition. In our identification results, CP022110 is classified as a chromosome, CP022111 is classified as a chromid, and CP022112 and CP022113 are classified as plasmids. Therefore, in our future update plans, we will expand the databases for Rep proteins and core genes, particularly the core gene database encompassing diverse habitats, to improve the tool's true positive rate (TPR).

## Analysis of chromids in MAGs

We input 92,143 MAGs from the human gut microbiome into Chromid-Finder for identification. With the tetranucleotide relative abundance distance set to 1.6, we identified 27,995 clusters and 4,682 chromid sequences. The phenomenon of a significantly larger number of clusters than chromids is attributed to the presence of 2,505 human gut species among the 92,143 MAGs (11). The center sequences of the clusters (i.e., potential bacterial main chromosomes) belong to the same gut species, resulting in the same chromid appearing in different clusters.

Subsequently, we removed samples that did not simultaneously contain chromids and cluster center sequences, yielding 3,065 samples that detected potential chromid-carrying bacteria. We analyzed the relationship between the distribution of chromid-carrying bacteria and the host's living environment, including host age, health status (disease presence, antibiotic treatment, disease type), and geographical location. The results are illustrated in Fig. 9a. The distribution of chromid-carrying bacteria showed a statistically significant relationship with the host's age, health status, and geographical location. Notably, healthy hosts exhibited a higher than expected abundance of chromid-carrying bacteria; the use of antibiotics reduced their distribution. Additionally, there were more chromid-carrying bacteria in adolescents and adults, while the distribution in infants was significantly lower.

To visually represent the impact of geographical location and disease type on the distribution of chromid-carrying bacteria, we calculated the standardized residuals for the relevant indicators, as shown in Fig. S6. Regarding geographical factors, the differences in the distribution of chromid-carrying bacteria were most pronounced between Asia and South America. The distribution in Asia was significantly higher than expected, while in South America, it was significantly lower, potentially linked to dietary and lifestyle differences between the two continents. In terms of disease types, the most notable influences were observed in premature infants and type 2 diabetes (T2D). The distribution of chromid-carrying bacteria in T2D was significantly increased compared to expectations, while in premature infants, the distribution was significantly decreased. The impact of T2D on chromid-carrying bacteria distribution may relate to the dietary habits of the host. In contrast, premature infants present challenges in determining whether the age of the infant or the condition itself influences the distribution of chromid-carrying bacteria.

We utilized the data provided by Alexandre Almeida et al. in the literature for analysis. From the predicted protein-coding sequences, we used InterProScan (18) to generate annotations, which were translated to 1,199 genome properties (19, 20) (GPs) and 115 metagenomics gene ontology (21, 22) (GO) slim terms, a summarized classification of GO annotations from metagenomic data (23). Each GP, a functional attribute predicted to be encoded in a genome, was determined to be present, partially present, or absent, depending on the number of proteins that were detected to be involved in that property. Globally, by analyzing the repertoire of GPs according to the taxonomic composition, we observed a good separation by phylum (analysis of similarities [ANOSIM] $R = 0.42$, $P < 0.001$), with the *Bacteroidota and Pseudomonadota* taxa in particular displaying very distinctive functional profiles (Fig. S7a). We further investigated the separation between the chromid-carrying bacteria and non-chromid-carrying bacteria genomes within each phylum, which revealed a strong differentiation among *Mycoplasmatota* (ANOSIM $R \geq 0.30$; Fig. S7b).

As shown in Fig. 9b, although only *Mycoplasmatota* exhibited strong differentiation in the principal component analysis (PCA) based on GPs, the GO functional analysis revealed statistically significant differences in *Actinobacteriota, Bacteroidota, Bacillota,* and *Mycoplasmatota* (Wilcoxon rank-sum test, adjusted $P < 0.05$, effect size >0.2), with particularly pronounced differences in *Mycoplasmatota*. We observed that chromid-carrying bacteria genomes are enriched with functional genes related to RNA metabolism and translation processes. The presence of chromids may enhance protein production in bacteria, thereby aiding their adaptation to environmental conditions. As George C. diCenzo et al. observed, chromids are biased toward environmental adaptation, with the functional categories enriched on chromids being similarly over-represented on the chromosomes of environmental genera (24). A notable example is found in the marine bacterium *Marinovum algicola*, where its chromid carries a highly expressed flagellum gene cluster essential for swimming motility, a critical survival trait in aquatic environments (25). Overall, these data indicate that the chromid-carrying bacteria species described here possess specific functions, while also enhancing our understanding of the biological traits of chromid-carrying bacteria.

## DISCUSSION

By establishing a workflow for screening chromids in complete bacterial genomes, we have demonstrated that chromid-carrying bacteria are found in various types of bacteria across the globe. By discussing the sizes of chromosomes and chromids in chromid-carrying bacteria, we propose that the hypothesis of the plasmid origin of chromids (26, 27) may be correct. This is supported by the observation that the average chromosome size in chromid-carrying bacteria is smaller than that in bacteria with plasmids or megaplasmids, while the average size of chromids exceeds that of megaplasmids. This suggests that, over long-term evolution, plasmids or megaplasmids may have undergone genetic recombination with chromosomes, acquiring core genes essential for bacterial survival. These core genes are subsequently retained and vertically transmitted to descendant bacterial cells. Consequently, chromids have assumed certain chromosomal functions, contributing to the maintenance of fundamental bacterial viability and becoming an indispensable component of the bacterial genome.

Our study demonstrates the critical roles that chromids play in bacterial respiratory chains and antiviral defense mechanisms. An intriguing finding emerged from analyzing the top ten most abundant anti-phage systems on bacterial chromosomes, chromids, and megaplasmids: Excluding systems shared by all three replicon types or unique to individual types, the anti-phage systems found on chromids show overlap with those found on both bacterial chromosomes and megaplasmids. In contrast, no such overlap exists directly between the systems specific to bacterial chromosomes and those specific to megaplasmids. This pattern underscores that chromids represent a distinct class of replicon, intermediate between bacterial chromosomes and megaplasmids. Furthermore, it provides additional support for the plasmid-origin hypothesis of chromids.

The human gut microbiota is one of the most studied microbial environments, but technical and practical constraints hinder our ability to isolate and sequence every constituent species. Metagenomic methods provide access to the uncultured microbial diversity. Here, the bioinformatics software we developed, Chromid-Finder, expands the current potential for chromid identification and characterization from metagenomic sequences. After testing with data sets, our software demonstrates excellent performance, accurately identifying chromid sequences from metagenomic data.

By annotating MAGs with Chromid-Finder, we found that the distribution of chromid-carrying bacteria is closely related to the host's living environment and health status. Compared to non-chromid bacteria, chromid-carrying bacteria within the same phylum also display unique functions. Our findings show that our workflow helps to uncover how chromid bacteria adapt to their environments, such as how they drive unique metabolic functions during infection or human gut dysbiosis.

As the first software capable of identifying chromids in metagenomes, there are still many areas for improvement. For instance, based on the characteristics of chromid-carrying bacteria and chromids, we plan to develop specific marker genes for different genera. Similarly, we aim to develop different core genes for various habitats, such as marine, volcanic, or animal gut environments. We will also continuously enrich our

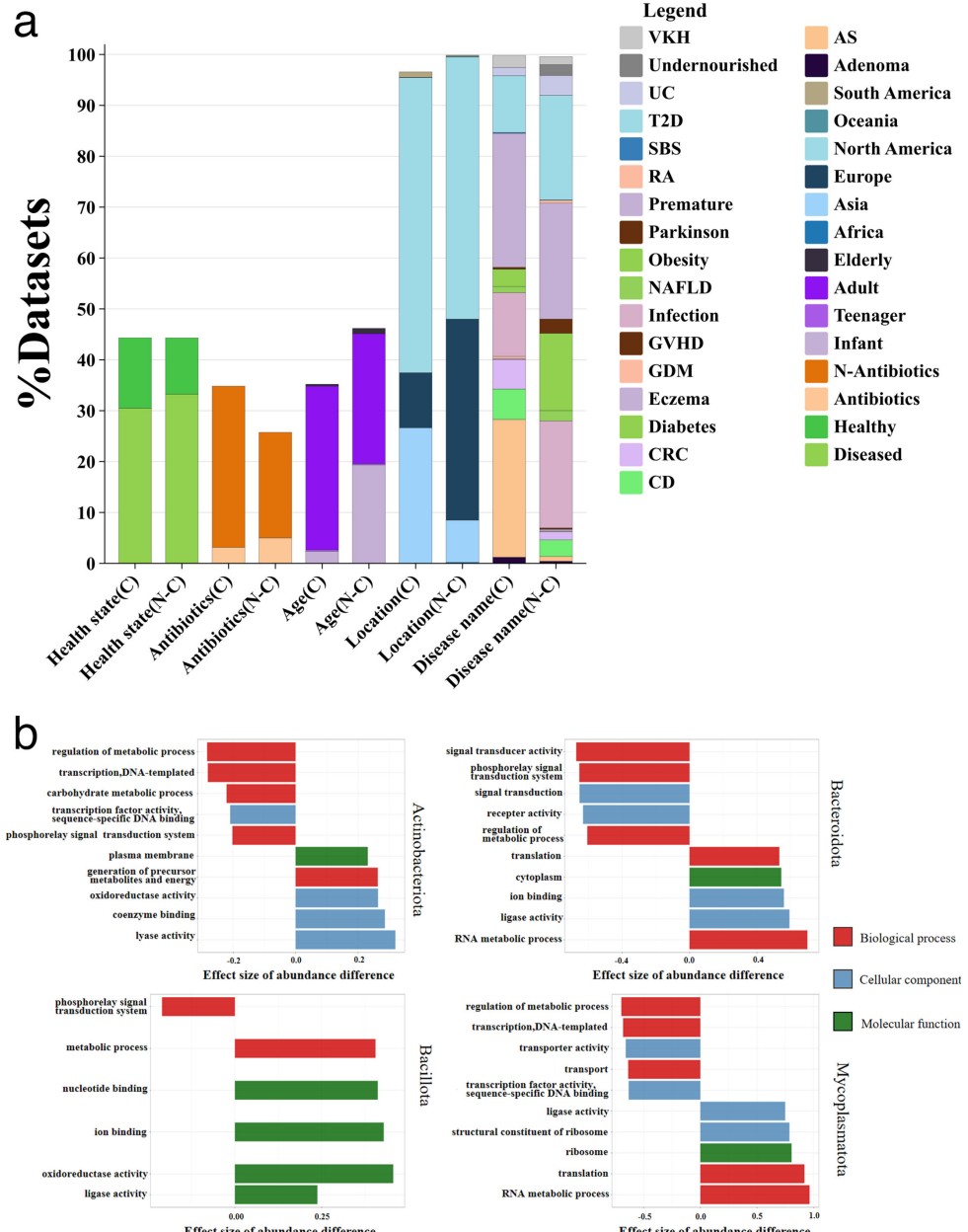

**FIG 9** Analysis of chromids in MAGs. (a) Percentages of chromid-carrying bacteria samples (C) and non-chromid-carrying bacteria samples (N-C) within the metagenomic data set (chi-square analysis; *P*-values all less than 0.001). (b) GO functions are differentially abundant between the chromid-carrying bacteria and non-chromid-carrying bacteria genomes from Actinobacteria, Bacteroidetes, Firmicutes, and Tenericutes. The five functions with the highest and lowest effect sizes of abundance difference with a false discovery rate <5% are represented. A positive effect size denotes overrepresentation in the chromid-carrying bacteria genomes. AS, ankylosing spondylitis; CD, Crohn's disease; CRC, colorectal cancer; GDM, gestational diabetes mellitus; GVHD, graft-versus-host disease; NAFLD, non-alcoholic fatty liver disease; RA, rheumatoid arthritis; SBS, short bowel syndrome; T2D, type 2 diabetes; UC, ulcerative colitis; VKH, Vogt-Koyanagi-Harada disease. Obesity was defined as a body mass index >30 kg m$^{-2}$. Infant: 0–3 years; teenager: 13–18 years; adult: 19–64 years; elderly: >64 years.

HMM database related to plasmid replication and segregation to improve the recall of chromid sequences in metagenomic data. Along with these improvements, with the development of artificial intelligence and big data models, we may also explore using machine learning techniques to create software with similar functions.

In summary, we hope that the release of Chromid-Finder will not only have a significant impact in the field of bacterial genomics but also bring unprecedented power to areas such as microbial ecology, biotechnology, and the investigation of the relationship between microbiota and human health and disease.

## MATERIALS AND METHODS

### Data source

i. We downloaded 36,322 complete bacterial genome sequences from the NCBI RefSeq database (10) using the ncbi-genome-download tool (28) (http://www.ncbi.nlm.nih.gov/refseq/, last accessed: 28 November 2023). The data included nucleotide sequence (fna) files and GenBank (gbff) files.
ii. Chromid genome sequence data were collected from 39 bacterial genomes across 27 publications, comprising a total of 156 sequences: 39 chromosome sequences, 45 chromid sequences, and 72 plasmid sequences (9, 17, 29–53). The accession numbers and corresponding references are listed in Table S2.
iii. We used 92,143 MAGs from the human gut microbiome collected by Alexandre Almeida et al. (11).

### Construction of hidden Markov model database for chromid identification

We constructed an HMM database consisting of 53 plasmid replication system (Rep) specific HMM files, 44 plasmid partitioning system (Par) specific HMM files, and 136 bacterial core gene HMM files, along with two additional HMM files for DnaA and its homologs sourced from the Kyoto Encyclopedia of Genes and Genomes (KEGG) database.

Rep-specific HMM files were as follows.

i. Fifteen files were retrieved from the Pfam database (54) (version 36.0, 2023.9, https://pfam.xfam.org/).
ii. Fourteen files were extracted from the DPR database by Dr. Mark Osborn (archived in 2009), followed by MAFFT (v7.525) (55) alignment and hmmbuild commands in HMMER (v3.4) (56) to create the HMMs.
iii. Twenty-four files were generated following the Lanza method (57), clustering plasmid replication initiation proteins from UniProt (58) using CD-HIT (v4.8.1) at a 40% threshold.

Par-specific HMM files were as follows.

i. Forty files were obtained from Pfam, pVOG (59), TIGRFAM (60), and eggNOG (61).
ii. Four additional files were based on Gerdes' research (62), processed using MAFFT and hmmbuild.

Core gene HMM files were as follows. One hundred twenty files were integrated from GTDB-Tk (v2) (63) and 81 from UBCG2 (64), yielding 136 unique HMMs after removing redundancies.

## Screening for bacterial genomes with chromids

i. Among the 36,322 downloaded genomes, 17,236 with two or more nucleotide sequences were filtered for further analysis.
ii. The longest sequence in each genome was designated as the chromosome, while the remaining sequences were considered secondary replicons (5).
iii. GC content differences between chromosomes and secondary replicons were calculated, with putative chromids identified when the difference was within ±1% (6).
iv. Gene prediction was performed using Prodigal (v2.6.3) (65).
v. HMMER's hmmsearch command was used to align sequences against our HMM database. A secondary replicon was classified as a chromid if it contained proteins related to plasmid replication, plasmid partitioning, and core genes (6).
vi. The classify_wf command in GTDB-Tk was used to align multiple sequences from bacterial chromosomes containing putative chromids, and the infer command was used to construct phylogenetic trees from the aligned sequences. The results were visualized online using tvBOT (https://www.chiplot.online/tvbot.html) (66).
vii. Sankey plots for bacterial classification were generated using ChiPlot (https://www.chiplot.online/).

## Analysis of chromid characteristics

tRNAs and rRNAs in chromid genomes were predicted using tRNAscan-SE (v2.0.11) (67) and barrnap (v0.9). Custom Python scripts were employed to calculate dinucleotide relative abundance distances (5) and analyze chromid characteristics, including genome size, GC content, and the number of Rep proteins, Par proteins, core genes, tRNAs, and rRNAs. Bar charts and heatmaps were generated with ChiPlot.

## Comparative analysis of bacterial respiratory chain genes

KEGG pathways were annotated using KofamScan (v1.3.0) (68). Custom Python scripts were developed to parse KEGG pathway hierarchies. Following the method described by Kaila (14), respiratory chain models and binary plots for bacteria containing putative chromids were generated.

## Identification and analysis of bacterial defense systems

Defense systems and CRISPR-Cas systems in bacterial genomes were predicted using Defense-Finder (v1.2.0) (69) and CRISPRCasFinder (v4.3.2) (70), both based on the MacSyFinder platform. Spacer sequences extracted from CRISPR loci were aligned against the phage sequences in the NCBI viral database using BLASTN-short (71). Matches with an alignment length greater than 24 bp and less than one mismatch were retained to identify phages infecting the target bacteria.

## Global distribution of bacteria with chromids

The geographical coordinates and country information provided in the NCBI RefSeq database were used to map the global distribution of bacteria containing chromids. Data with unclear or unavailable information were excluded, and the global distribution map was generated using ChiPlot.

## Setting parameters for tetranucleotide relative abundance distance

We utilized a custom Python script to calculate the tetranucleotide relative abundance distance between the 1,163 chromids and their corresponding main chromosomes from our previous research. We used ChiPlot to generate bar charts and, based on a comprehensive assessment of precision and false positive rate (FPR), recommended a reference value of 1.6.

## Testing Chromid-Finder

Using the two test data sets from Data Source (1) and (2), we removed the correspondence and input them as independent genomic data into Chromid-Finder: one test involved 62,850 sequences, including 1,163 positive data, and the other involved 156 sequences, including 45 positive data. We evaluated Chromid-Finder's performance based on the following formulas:

Accuracy: Acc = (TP + TN)/(P + N),

Precision: TP/(TP + FP) = TP/P,

True positive rate (TPR)/sensitivity/recall: TP/(TP + FN) = TP/T,

False positive rate (FPR): FPR = FP/(FP + TN) = FP/F.

## Analysis of metagenome-assembled genomes from the human gut microbiome

i. We utilized Chromid-Finder to identify chromid sequences from 92,143 MAGs derived from the human gut microbiome. Using a tetranucleotide relative abundance distance threshold of 1.6, we obtained a total of 55,624 clusters (each cluster containing a potential chromosome sequence along with one or more corresponding chromid sequences) and 4,740 chromid sequences.

ii. These 92,143 MAGs were sourced from 13,133 human gut metagenomic data sets. We eliminated bacterial chromosome sequences and chromid sequence data that did not originate from the same sample. By combining this data with 2,505 identified human gut species from the literature, we ultimately identified 3,048 samples that potentially harbor chromid-carrying bacteria, comprising 573 species with chromids and 1,932 species without chromids.

iii. We conducted a statistical analysis comparing samples with and without detected chromid-carrying bacteria, assessing the percentages of host age, health status (including whether the host is diseased, has undergone antibiotic treatment, and the type of disease), and geographic location in the data set. We performed a chi-squared analysis and standardized residuals detection using the chi2_contingency and residuals functions in Python to determine whether these factors are statistically associated with the distribution of chromid-carrying bacteria.

iv. We performed a functional comparative analysis between species that possess chromids and those that do not. This analysis was based on 1,199 GPs and 115 metagenomics GO data corresponding to the 2,505 human gut species provided in the literature (26). Differential abundance analysis of GO slim and GPs frequencies between the chromid-carrying bacteria and non-chromid-carrying bacteria genomes was performed with the compositional data analysis tool ALDEx2 (72). The aldex.clr function was used with 128 Monte Carlo instances sampled from a Dirichlet distribution to generate a distribution of probabilities for each GO slim term consistent with the observed data. These were subsequently converted to distributions of log ratios to account for the compositional nature of the data. The aldex.effect function was used to calculate the expected value of the difference between distributions of each group (median log2 difference), the expected value of the pooled group variance (median log2 dispersion), and the standardized effect sizes on the abundance difference of each GO classification. Lastly, the aldex.ttest was used to perform non-parametric Wilcoxon rank-sum tests on the GO frequencies between the two test groups (Chromid and N_chromid). GPs, classified as "yes," "no," and "partial," were converted to 2, 0, and 1, respectively, and those more prevalent specifically among the chromid-carrying bacteria genomes were detected with a two-tailed $\chi^2$ test. The expected $P$ values from all the statistical tests were corrected for multiple testing with the Benjamini–Hochberg method. A PCA was carried out on the GP distributions of the non-chromid-carrying bacteria and chromid-carrying bacteria genomes, using the

FactorMineR (73) package. Separation according to phylum and genome type was assessed with the ANOSIM test based on the Gower distances between the GP profiles.

## ACKNOWLEDGMENTS

Thanks go to all of the contributors to this work.

This work was supported financially by the Science and Technology Program of Xinjiang Production and Construction Corps (2024AB050), the Third Xinjiang Scientific Expedition Program (2022xjkk1202 and 2022xjkk0804), the Bingtuan Science and Technology Project (NYHXGG2023AA101), the Tianshan Talent Project (2022TSYCCX0125 and 2023TSYCJU0010), and the Tianshan Talent innovation team (2023TSYCTD0021).

## AUTHOR AFFILIATIONS

[1]College of Life Sciences, Shihezi University, Shihezi, Xinjiang, China
[2]Opthalmic Center, Xinjiang 474 Hospital, Urumqi, Xinjiang, China
[3]State Key Laboratory of Sheep Genetic Improvement and Healthy Production, Xinjiang Academy of Agricultural and Reclamation Sciences, Shihezi, Xinjiang, China
[4]College of Veterinary Medicine, Xinjiang Agriculture University, Urumqi, Xinjiang, China

## AUTHOR ORCIDs

Haoyu Liu http://orcid.org/0009-0003-4417-2816
Shengwei Hu http://orcid.org/0000-0001-8849-265X

## FUNDING

| Funder | Grant(s) | Author(s) |
| --- | --- | --- |
| Science and Technology Bureau of Xinjiang Production and Construction Corps | 2024AB050 | Haoyu Liu |
| | | Jia Sun |
| | | JuanJuan Si |
| | | Yi Liao |
| | | Jiaqi Bai |
| | | Xia Li |
| | | Limin Wang |
| | | Kuojun Cai |
| | | Wei Ni |
| | | Ping Zhou |
| | | Shengwei Hu |
| Science and Technology Bureau of Xinjiang Production and Construction Corps | NYHXGG2023AA101 | Haoyu Liu |
| | | Jia Sun |
| | | JuanJuan Si |
| | | Yi Liao |
| | | Jiaqi Bai |
| | | Xia Li |
| | | Limin Wang |
| | | Kuojun Cai |
| | | Wei Ni |
| | | Ping Zhou |
| | | Shengwei Hu |

| Funder | Grant(s) | Author(s) |
|---|---|---|
| Science and Technology Bureau of Xinjiang Production and Construction Corps | 2022TSYCCX0125 | Haoyu Liu<br>Jia Sun<br>JuanJuan Si<br>Yi Liao<br>Jiaqi Bai<br>Xia Li<br>Limin Wang<br>Kuojun Cai<br>Wei Ni<br>Ping Zhou<br>Shengwei Hu |
| Science and Technology Bureau of Xinjiang Production and Construction Corps | 2023TSYCJU0010 | Haoyu Liu<br>Jia Sun<br>JuanJuan Si<br>Yi Liao<br>Jiaqi Bai<br>Xia Li<br>Limin Wang<br>Kuojun Cai<br>Wei Ni<br>Ping Zhou<br>Shengwei Hu |
| Science and Technology Bureau of Xinjiang Production and Construction Corps | 2023TSYCTD0021 | Haoyu Liu<br>Jia Sun<br>JuanJuan Si<br>Yi Liao<br>Jiaqi Bai<br>Xia Li<br>Limin Wang<br>Kuojun Cai<br>Wei Ni<br>Ping Zhou<br>Shengwei Hu |
| Third Xinjiang Scientific Expedition Program | 2022xjkk1202, 2022xjkk0804 | Haoyu Liu<br>Jia Sun<br>JuanJuan Si<br>Yi Liao<br>Jiaqi Bai<br>Xia Li<br>Limin Wang<br>Kuojun Cai<br>Wei Ni<br>Ping Zhou<br>Shengwei Hu |

## AUTHOR CONTRIBUTIONS

Haoyu Liu, Conceptualization, Data curation, Methodology, Resources, Software, Validation, Visualization, Writing – original draft | Jia Sun, Funding acquisition, Project administration, Resources | JuanJuan Si, Formal analysis, Writing – review and editing | Yi Liao, Software, Validation | Jiaqi Bai, Formal analysis, Writing – review and editing | Xia Li, Data curation, Supervision | Limin Wang, Funding acquisition, Supervision | Kuojun Cai, Project administration, Supervision | Wei Ni, Formal analysis, Supervision | Ping Zhou, Funding acquisition, Supervision | Shengwei Hu, Project administration, Resources, Supervision, Writing – review and editing

## DATA AVAILABILITY

Data can be obtained from the corresponding author upon reasonable request or from the included supplemental information files. Chromid-Finder is implemented in Python, and all scripts and associated files are freely available at https://github.com/China-LHY/Chromid-Finder.

## ADDITIONAL FILES

The following material is available online.

### Supplemental Material

**Supplemental figures (mSystems00175-25-S0001.pdf).** Fig. S1–S7.
**Process document (mSystems00175-25-S0002.xlsx).** The information for 36,322 bacterial genomes along with the screening process files.
**Table S1 (mSystems00175-25-S0003.xlsx).** Information on 1,104 chromids and their corresponding bacterial chromosomes.
**Table S2 (mSystems00175-25-S0004.xlsx).** The ratio of chromid-carrying strain genomes to the total number of complete genomes sequenced within each bacterial genus.
**Table S3 (mSystems00175-25-S0005.xlsx).** Chromid-carrying bacterial genomes with available isolation source.
**Table S4 (mSystems00175-25-S0006.xlsx).** Relevant information about the 39 bacterial genomes in test data set 2.

### Open Peer Review

**PEER REVIEW HISTORY (review-history.pdf).** An accounting of the reviewer comments and feedback.

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
