## [Reviewer comments · mSystems]

Unexplored diversity and potential functions of extra-chromosomal elements

Haoyu Liu, Jia Sun, Juanjuan Si, Yi Liao, Jiaqi Bai, Xia Li, Limin Wang, Kuojun Cai, Wei Ni, Ping Zhou, and Shengwei Hu

Corresponding Author(s): Shengwei Hu, Shihezi University College of Life Science

Review Timeline:

Submission Date:	March 13, 2025
Editorial Decision:	May 9, 2025
Revision Received:	June 16, 2025
Editorial Decision:	June 20, 2025
Revision Received:	June 24, 2025
Accepted:	June 27, 2025

Editor: Juliette Hayer

Reviewer(s): The reviewers have opted to remain anonymous.

Transaction Report:

DOI: <https://doi.org/10.1128/msystems.00175-25>

Re: mSystems00175-25 (**Unexplored diversity and potential functions of extra-chromosomal elements**)

Dear Dr. Shengwei Hu:

Thank you for the work done during the revision period, you have addressed most of the comments from reviewers #1 and #2. Unfortunately, reviewer #2 was not available to evaluate the revised manuscript, so I asked reviewer #3 to assess this revised version.

Reviewer #3 has some comments and raised questions that are interesting and I would like you to take this into consideration for an additional revision of your manuscript.

Revision Guidelines

Sincerely,
Juliette Hayer
Editor
mSystems

Reviewer #1 (Comments for the Author):

"I thank the authors for having seriously considered and addressed most of my comments.

Reviewer #3 (Comments for the Author):

This is my first time reviewing the manuscript "Unexplored diversity and potential functions of extra-chromosomal elements" by Liu et al. This manuscript reports the results of a screen of over 30,000 bacterial genomes for chromids and an analysis of the gene content of some of them. The manuscript additionally reports the development of a computational tool, Chromid-Finder, for the automated identification of chromids from genome sequences. Developing a tool for identification of chromids is an interesting idea. However, I have several suggestions and comments for the Authors consideration.

Major comments

I agree with previous Reviewer 2 that the low TPR score in the second test is a concern.

The first test assumes that the chromids the Authors identified earlier in the manuscript are all true chromids, which is not a valid assumption without experimental validation of the essential nature of these replicons (as the key defining feature of a chromid is that it is essential for cell viability). In addition, for the first test, the Authors first identify putative chromids using a set of criteria, and then test their software on this same dataset, where the software is using the same criteria as the first search used. In other words, this is not an independent test of the accuracy of the software, but rather a confirmation that it correctly implements the desired steps.

In contrast, the second set is a curated set of genomes known to contain chromids, and thus is the only true test of the software in this manuscript. Given the low TPR score in this test, I recommend the Authors work on improving this score (using the approaches they suggest in the manuscript) prior to publishing the tool

I agree with previous Reviewer 1 that more testing of thresholds would be valuable. While the Authors have added some of this data for the tetranucleotide relative abundance distance, I think it would be equally important to look at the effect of the GC content difference threshold.

I also agree with previous Reviewer 1 that some of the data analyses are incomplete and that it is difficult to know which results are specific to a limited number of chromids and which are generalizable. For example, the Authors describe five examples of how chromids can contribute to bacterial respiratory chains. But it is not clear to me what percentage of chromids carry such genes or fall into each category, and whether this differs from megaplasms.

MAGs generally do not have fully completed genome assemblies, meaning that the chromid (and chromosome) of MAGs are likely to be split across multiple chromids. How does Chromid-Finder deal with this situation?

Other comments

How do chromids compare to megaplasms in terms of contributions to antiviral defense?

I would like to see a list of all ~36,000 genomes analyzed for chromids in this study. There are several organisms missing from the list of organisms with chromids that I would have expected to see, and thus I would like to see if these organisms were excluded from the analysis or if they are false negatives.

Response to Reviewers

Manuscript Title: Unexplored diversity and potential functions of extra-chromosomal elements

Manuscript ID: mSystems00175-25

Journal Name: mSystems

Dear Editor and Reviewers,

We sincerely appreciate the time and effort that the reviewers and the editor have invested in evaluating our manuscript. We are grateful for their insightful comments and constructive suggestions, which have helped us improve the quality and clarity of our work. Below, we provide detailed responses to each comment.

Reviewer #3

Comment 1: *I agree with previous Reviewer 2 that the low TPR score in the second test is a concern.*

Response: Thank you for your reminder. In the revised manuscript, we optimized the hmmsearch command (see Response 4). The True Positive Rate (TPR) value has increased from 51.11% to 73.33% (Table 2).

Comment 2: *The first test assumes that the chromids the Authors identified earlier in the manuscript are all true chromids, which is not a valid assumption without experimental validation of the essential nature of these replicons (as the key defining feature of a chromid is that it is essential for cell viability).*

Response: Indeed, you make a valid point. The chromids identified through bioinformatic analysis can only be referred to as putative chromids. We plan to experimentally validate some of the bacteria of interest in future work.

Comment 3: *In addition, for the first test, the Authors first identify putative chromids using a set of criteria, and then test their software on this same dataset, where the software is using the same criteria as the first search used. In other words, this is not an independent test of the accuracy of the software, but rather a confirmation that it correctly implements the desired steps.*

Response: Yes, we confirmed using test dataset 1 that the software correctly implements the required steps. Furthermore, compared to our initial workflow, our tool has relaxed the filtering criteria for Rep proteins, Par proteins, and core genes (see Response 4), and additionally incorporated HMM profiles related to bacterial chromosome replication origin specificity and the parameter of tetranucleotide relative abundance distance. This was done with the aim of testing whether the software can effectively function as a screening tool when faced with numerous independent replicon sequence inputs, rather than when comparing different replicons within the same genome. The results demonstrate that at a tetranucleotide relative abundance distance threshold of 1.6, we identified 1043 chromids out of 1163 putative chromids in test dataset 1, achieving a recall rate of 89.68%.

Comment 4: *In contrast, the second set is a curated set of genomes known to contain chromids, and thus is the only true test of the software in this manuscript. Given the low TPR score in this test, I recommend the Authors work on improving this score (using the approaches they suggest in the manuscript) prior to publishing the tool*

Response: Thank you for the reviewer's suggestion. We optimized the hmmsearch command in script part1.py. Through this implementation, we increased the True Positive Rate (TPR) from 51.11% to 73.33% (Table 2).

Original: `f"hmmsearch -Z 1 --noali --cut_ga --cpu 2 --domtblout {input_file}-core1.out databases/core1.hmm {input_file}.faa",`

`f"hmmsearch -Z 1 --noali --cut_ga --cpu 2 --domtblout {input_file}-core2.out databases/core2.hmm {input_file}.faa",`

`f"hmmsearch -Z 1 --noali --cut_ga --cpu 2 --domtblout {input_file}-par1.out databases/par1.hmm {input_file}.faa",`

```
f"hmmsearch -Z 1 --noali --cut_ga --cpu 2 --domtblout {input_file}-rep1.out databases/rep1.hmm  
{input_file}.faa",
```

```
Revised: f"hmmsearch -Z 1 --noali --domE 1e-5 --cpu 2 --domtblout {input_file}-core1.out databases/core1.hmm  
{input_file}.faa",
```

```
f"hmmsearch -Z 1 --noali --domE 1e-5 --cpu 2 --domtblout {input_file}-core2.out databases/core2.hmm  
{input_file}.faa",
```

```
f"hmmsearch -Z 1 --noali --domE 1e-5 --cpu 2 --domtblout {input_file}-par1.out databases/par1.hmm  
{input_file}.faa",
```

```
f"hmmsearch -Z 1 --noali --domE 1e-5 --cpu 2 --domtblout {input_file}-rep1.out databases/rep1.hmm  
{input_file}.faa",
```

keeping the Score ≥ 30 threshold unchanged. We also conducted a detailed analysis of the identification process (see Supplementary Table 4: Relevant information about the 39 bacterial genomes in test dataset 2). In summary, out of 45 chromids across these 39 bacterial genomes, we accurately identified 33 chromids (32+1). Among the remaining 12 chromids that were not identified:

10 lacked a core gene (no core gene, $n=10$),

7 lacked a Rep protein (no Rep, $n=7$),

and 5 had a difference in GC content exceeding 1% (difference in GC content $> 1\%$, $n=5$).

Furthermore, regarding GCF_000730165.1, the cited article asserts that there are three chromosomes in this genome, which conflicts with our definition. In our identification results:

CP022110 is classified as a chromosome,

CP022111 is classified as a chromid,

and CP022112 and CP022113 are classified as plasmids.

Therefore, in our future update plans, we will expand the databases for Rep proteins and core genes, particularly the core gene database encompassing diverse habitats, to improve the tool's True Positive Rate (TPR).

Comment 5: *I agree with previous Reviewer 1 that more testing of thresholds would be valuable. While the Authors have added some of this data for the tetranucleotide relative abundance distance, I think it would be equally important to look at the effect of the GC content difference threshold.*

Response: We appreciate the reviewer's caution. However, in this study, the GC content difference served as a fixed criterion ($\leq 1\%$), not as a tunable parameter. Therefore, we did not explore varying this threshold.

Comment 6: *I also agree with previous Reviewer 1 that some of the data analyses are incomplete and that it is difficult to know which results are specific to a limited number of chromids and which are generalizable. For example, the Authors describe five examples of how chromids can contribute to bacterial respiratory chains. But it is not clear to me what percentage of chromids carry such genes or fall into each category, and whether this differs from megaplasms.*

Response: Thank you for your valuable suggestion. We re-analyzed the presence of respiratory chain enzyme complex-related proteins on bacterial chromosomes and chromids. Additionally, referencing the standard proposed by George C. diCenzo et al. of 350 kb for defining megaplasms, we downloaded and analyzed 472 megaplasms originating from 1104 chromid-carrying bacterial genomes.

For the following three functional categories:

- (i) No involvement in the bacterial respiratory chain (e.g., chromid-3 in *Azospirillum brasilense* and the chromid in *Vibrio chagasii*).
- (ii) Independent encoding of all protein genes for a respiratory chain enzyme complex (e.g., chromid-1 in *Azospirillum brasilense* and the chromid in *Burkholderia pseudomallei*).
- (iii) Supporting the bacterial chromosome in synthesizing respiratory chain enzyme complexes (e.g., chromid-2 in *Azospirillum brasilense*, the chromid in *Paracoccus versutus*, and the chromid in *Vibrio chagasii*).

The distribution across replicon types was:

In bacterial chromosomes: (2 / 1104), (1080 / 1104), (22 / 1104)

In bacterial chromids: (144 / 1163), (521 / 1163), (489 / 1163)

In bacterial megaplasms: (233 / 472), (34 / 472), (205 / 472)

These results clearly demonstrate the distinct functional roles played by different replicon types within the bacterial respiratory chain.

Comment 7: *MAGs generally do not have fully completed genome assemblies, meaning that the chromid (and chromosome) of MAGs are likely to be split across multiple chromids. How does Chromid-Finder deal with this situation?*

Response: Indeed, your understanding is correct. Therefore, we require users to input a single FASTA file containing multiple sequences, ensuring that each sequence is relatively complete, and avoid situations where a sequence is composed of multiple fragments such as xx. bin1, xx. bin2. In future versions, we will consider integrating a de novo assembly and binning workflow.

Comment 8: *How do chromids compare to megaplasms in terms of contributions to antiviral defense?*

Response: Following the standard proposed by George C. diCenzo et al. of 350 kb for defining megaplasms, we downloaded and analyzed 472 megaplasms originating from 1104 chromid-carrying bacterial genomes.

Using the simple calculation method established in our previous work, the density of anti-phage systems on megaplasms was determined to be 1.296 systems per Mbp.

Simultaneously, we optimized Fig. 6a. The results demonstrate that chromids play a crucial role in supplementing the bacterial defense arsenal and enhancing the capacity to resist phage attacks. An intriguing finding emerged from analyzing the top ten most abundant anti-phage systems on bacterial chromosomes, chromids, and megaplasms: Excluding systems shared by all three replicon types or unique to individual types, the anti-phage systems found on chromids show overlap with those found on both bacterial chromosomes and megaplasms. In contrast, no such overlap exists directly between the systems specific to bacterial chromosomes and those specific to megaplasms. This

pattern underscores that chromids represent a distinct class of replicon, intermediate between bacterial chromosomes and megaplasmid. Furthermore, it provides additional support for the plasmid-origin hypothesis of chromids.

Original Fig. 6a:

Revised Fig. 6a:

Comment 9: *I would like to see a list of all ~36,000 genomes analyzed for chromids in this study. There are several organisms missing from the list of organisms with chromids that I would have expected to see, and thus I would like to see if these organisms were excluded from the analysis or if they are false negatives.*

Response: We have packaged the information for 36,322 bacterial genomes along with the screening process files into Process_document.xlsx, which we hope contains the required data.

Process_document.zip includes:

Pd-1: Species and NCBI accession numbers for 36,322 bacterial genomes.

Pd-2: Information for 17,236 bacterial genomes with multipartite genomes.

Pd-3: 6,614 bacterial genomes containing potential chromids (where the GC content difference between the 'secondary replicon' and chromosome was $\leq 1\%$).

Pd-4: 6,614 bacterial genomes containing potential chromids (derived from Pd-3 by removing sequences where the GC content difference between a 'secondary replicon' and chromosome exceeded 1%).

Pd-5: 6,614 bacterial genomes containing potential chromids (expanded from Pd-4 with additional annotations for Rep protein distribution, Par protein distribution, core gene distribution, and Dinucleotide Distance).

The final results can be found in Supplementary Table 1: Information on 1,163 chromids and their corresponding bacterial chromosomes.

Additionally, due to the optimization of the tool's commands, we performed re-screening on the 92,143 genomes derived from the human gut metagenome and re-plotted Fig. 8b, Fig. 9a, Fig. 9b, Extended Data Fig. S6, and Extended Data Fig. S7.

Original Fig. 8b

Revised Fig. 8b

Original Fig. 9a

Revised Fig. 9a

Original Fig. 9b

Revised Fig. 9b

Original Extended Data Fig. S6

Revised Extended Data Fig. S6

Original Extended Data Fig. S7

Revised Extended Data Fig. S7

Once again, we sincerely appreciate the reviewers' and the editor's constructive feedback. We believe that the revisions have significantly improved our manuscript. We hope that our responses address all concerns and that the revised manuscript is now suitable for publication in mSystems.

Best regards,

Dr. Shengwei Hu

College of Life Sciences,

Shihezi University, Xin Jiang, China

Tel: 0993-2058002

Fax: 0993-2058612

E-mail: hushengwei@163.com

Re: mSystems00175-25R1 (**Unexplored diversity and potential functions of extra-chromosomal elements**)

Dear Dr. Shengwei Hu:

Thank you, you have addressed and answered most of our concerns and comments and the manuscript is now almost ready for publication.

Just a few minors modifications to take into account before publication:

fig 6: spelling mistake at Plasmids

In Figure S7 and in the text (ex. line 315): please use the updated taxonomy phyla names. Firmicutes, Bacteroidetes... are now Bacillota and Bacteroidota for example. NCBI and GTDB have had these new names for 2 years now so it is appropriate to use them.

Fig 9a: the y axis is Datasets and not datadets I guess, please correct.

Revision Guidelines

Sincerely,
Juliette Hayer
Editor
mSystems

Response to Reviewers

Manuscript Title: Unexplored diversity and potential functions of extra-chromosomal elements

Manuscript ID: mSystems00175-25R1

Journal Name: mSystems

Dear Dr. Juliette Hayer and Reviewers,

Thank you very much for your positive evaluation of our revised manuscript and for providing the final minor comments to further improve the manuscript. We appreciate the time and effort you have dedicated to reviewing our work. We have carefully addressed all the points raised and believe the manuscript has been strengthened accordingly. Below is our point-by-point response to the comments.

Comments from the Editor:

Comment 1: *fig 6: spelling mistake at Plasmids*

Response: We sincerely apologize for this oversight. We have corrected this error in Figure 6.

Original Fig. 6a:

Revised Fig. 6a:

Comment 2: In Figure S7 and in the test (ex. line 315): please use the updated taxonomy phyla names. *Firmicutes*, *Bacteroidetes*... are now *Bacillota* and *Bacteroidota* for example. NCBI and GTDB have had these new names for 2 years now so it is appropriate to use them.

Response: Thank you for pointing this out. We have updated all the phylum names throughout the manuscript and supplementary materials.

Original Fig. 9b:

Revised Fig. 9b:

Extended Data Fig. S7

Revised Extended Data Fig. S7

Comment 3: *the y axis is Datasets and not datadets I guess, please correct.*

Response: We apologize for this typographical error. The y-axis label in Figure 9a has been corrected from "Datadets" to "Datasets" in the revised figure file (Figure 9a).

Original Fig. 9a:

Revised Fig. 9a:

Once again, we sincerely appreciate the reviewers' and the editor's constructive feedback. We believe that the revisions have significantly improved our manuscript. We hope that our responses address all concerns and that the revised manuscript is now suitable for publication in mSystems.

Best regards,

Dr. Shengwei Hu

College of Life Sciences,

Shihezi University, Xin Jiang, China

Tel: 0993-2058002

Fax: 0993-2058612

E-mail: hushengwei@163.com

Re: mSystems00175-25R2 (**Unexplored diversity and potential functions of extra-chromosomal elements**)

Dear Dr. Shengwei Hu:

Your manuscript has been accepted, and I am forwarding it to the ASM production staff for publication. Your paper will first be checked to make sure all elements meet the technical requirements. ASM staff will contact you if anything needs to be revised before copyediting and production can begin. Otherwise, you will be notified when your proofs are ready to be viewed.

Sincerely,
Juliette Hayer
Editor
mSystems